# Urban Wildlife Crisis: Australian Silver Gull Is a Bystander Host to Widespread Clinical Antibiotic Resistance

Ethan R. Wyrsch,[a] Kristina Nesporova,[b,c] Hassan Tarabai,[b] Ivana Jamborova,[b] Ibrahim Bitar,[d] Ivan Literak,[b,c] Monika Dolejska,[b,c,d,e] Steven P. Djordjevic[a]

[a]Australian Institute for Microbiology & Infection, University of Technology Sydney, Ultimo, New South Wales, Australia
[b]CEITEC VETUNI, University of Veterinary Sciences Brno, Brno, Czech Republic
[c]Department of Biology and Wildlife Diseases, Faculty of Veterinary Hygiene and Ecology, University of Veterinary Sciences Brno, Brno, Czech Republic
[d]Faculty of Medicine, Biomedical Center, Charles University, Pilsen, Czech Republic
[e]Department of Clinical Microbiology and Immunology, Institute of Laboratory Medicine, The University Hospital Brno, Brno, Czech Republic

**ABSTRACT** The Australian silver gull is an urban-adapted species that frequents anthropogenic waste sites. The enterobacterial flora of synanthropic birds often carries antibiotic resistance genes. Whole-genome sequence analyses of 425 *Escherichia coli* isolates from cloacal swabs of chicks inhabiting three coastal sites in New South Wales, Australia, cultured on media supplemented with meropenem, cefotaxime, or ciprofloxacin are reported. Phylogenetically, over 170 antibiotic-resistant lineages from 96 sequence types (STs) representing all major phylogroups were identified. Remarkably, 25 STs hosted the carbapenemase gene $bla_{IMP-4}$, sourced only from Five Islands. Class 1 integrons carrying $bla_{IMP}$ and $bla_{OXA}$ alongside $bla_{CTX-M}$ and $qnrS$ were notable. Multiple plasmid types mobilized $bla_{IMP-4}$ and $bla_{OXA-1}$, and 121 isolates (28%) carried either a ColV-like (18%) or a pUTI89-like (10%) F virulence plasmid. Phylogenetic comparisons to human isolates provided evidence of interspecies transmission. Our study underscores the importance of bystander species in the transmission of antibiotic-resistant and pathogenic *E. coli*.

**IMPORTANCE** By compiling various genomic and phenotypic data sets, we have provided one of the most comprehensive genomic studies of *Escherichia coli* isolates from the Australian silver gull, on media containing clinically relevant antibiotics. The analysis of genetic structures capturing antimicrobial resistance genes across three gull breeding colonies in New South Wales, Australia, and comparisons to clinical data have revealed a range of trackable genetic signatures that highlight the broad distribution of clinical antimicrobial resistance in more than 170 different lineages of *E. coli*. Conserved truncation sizes of the class 1 integrase gene, a key component of multiple-drug resistance structures in the *Enterobacteriaceae*, represent unique deletion events that are helping to link seemingly disparate isolates and highlight epidemiologically relevant data between wildlife and clinical sources. Notably, only the most anthropogenically affected of the three sites (Five Islands) was observed to host carbapenem resistance, indicating a potential reservoir among the sites sampled.

**KEYWORDS** AMR, *Escherichia coli*, genomics, wildlife

Multiple-drug-resistant (MDR) infections, particularly those caused by the *Enterobacteriaceae*, are among the most threatening infections that directly impact human health (1). The incidence of MDR *Escherichia coli* in the healthy human gut, particularly lineages resistant to extended-spectrum $\beta$-lactams, has risen rapidly over the past 30 years (2, 3), primarily due to mobile genetic elements (MGEs) facilitating horizontal gene transfer. Horizontally acquired DNA often carries antimicrobial resistance genes (ARGs), metal resistance genes, virulence genes, or combinations of all three. The

Address correspondence to Steven P. Djordjevic, Steven.Djordjevic@uts.edu.au.

The authors declare no conflict of interest.

transfer of genetic material facilitates new combinations of MGEs and genes as well as new bacterial hosts for genes.

Eating fresh produce and retail meats and interacting with companion animals, livestock, recreational waters, and wildlife can introduce ARGs harbored by commensal bacteria and emergent pathogens into the human gut. This is significant because many infections caused by MDR bacteria are community acquired (4) or have an animal origin (5). These factors support the hypothesis that in the gut microbiome of animals, gene and bacterial host combinations are significant targets in efforts to prevent antimicrobial resistance (AMR) and pathogen evolution.

Wild birds, implicated as vectors of antibiotic resistance internationally (6, 7), are colonized by MDR and virulent lineages of *E. coli* and other *Enterobacteriaceae* that pose a threat to human health (8–13). Synanthropic birds, such as the Australian silver gull (*Chroicocephalus novaehollandiae*), carry multiply drug-resistant pandemic *E. coli* isolates, including globally dispersed extraintestinal pathogenic *E. coli* (ExPEC) lineages of sequence type 131 (ST131), ST69, ST10, ST1193, and ST38 (8, 9). The feeding behavior of the silver gull affords opportunities for it to acquire diverse *Enterobacteriaceae* from municipal dump sites, wastewater operations, and agricultural manure (14). Two large studies of *E. coli* isolated from Australian silver gulls (562 fecal samples and 284 cloacal samples) identified *E. coli* isolates resistant to critically important antimicrobials (CIAs) (8, 9). This phenomenon has been observed despite Australia having a long history of strict regulatory controls over antimicrobial usage in food production. Our understanding of the role that synanthropic birds play in the ecology and evolution of AMR and particularly pathogen emergence remains poor.

Previous studies of gull *E. coli* isolates have performed sampling at sites associated with human activity, e.g., where gulls congregate on beaches (9) and an island close to Australia's populated coast where gulls are known to feed almost entirely on human refuse (8). These studies have described (i) a rich diversity of drug-resistant *E. coli* lineages, (ii) complete sequences of resistance plasmids carrying genes encoding resistance to CIAs and evidence of their mobility (11), and (iii) emerging MDR (12). Evidence of nearly clonal *E. coli* isolates from gulls and humans is suggestive of interspecies transmission. However, these studies are limited to selected *E. coli* sequence types or have a focus on capturing data at the level of multilocus sequence typing (MLST) and genotyping for antimicrobial resistance and virulence genes. Both sampling of more pristine regions, ecologically less impacted by anthropogenic activity, and detailed analyses of plasmid and other MGE contents are notable omissions.

Plasmids are important purveyors of accessory genes that provide a competitive advantage in habitat colonization and resistance to antimicrobials. Two plasmid groups gaining attention for their appearance in *E. coli* isolates from diverse source environments, notably carried by pathogenic clade B ST131 lineages (15, 16), are distinct F plasmid subtypes known as colicin V (ColV) plasmids (17) and pUTI89 and related (pUTI89-like) plasmids (18, 19). Both subtypes harbor virulence-associated genes (VAGs) contributing to extraintestinal pathogenesis in *E. coli*. ColV plasmids carry a range of iron acquisition systems (*iutA*, *iucABCD*, *iroBCDN*, and *sitABCD*) and the additional effectors *iss*, *ompT*, *hlyF*, and *etsABC*, which have been identified on several F plasmid subtypes, including F2, F18, and F24 (17, 20). ColV-like plasmids are also a feature of diverse avian-pathogenic *E. coli* (APEC) lineages that cause colibacillosis and other systemic afflictions in commercial poultry (21). pUTI89-like plasmids are characterized by the replicon sequence types (RSTs) F29:A$^-$B10 and F2:A$^-$B10 and carry three to four separate virulence-associated regions, one of which includes the enterotoxin TieB (*senB*) (18). F plasmids have also shaped the evolution of dominant pandemic ST131 clade C (C1 F1:A2:B20 and C2 F2:A1:B–) lineages (22).

Class 1 integrons are important genetic elements for the evolutionary dynamics of antibiotic resistance. These elements reside on a variety of mobile genetic elements, can capture and express diverse resistance gene cassettes (23), and are important components in the evolution of complex resistance regions (CRRs) (24, 25). Diverse resistance genes, including metal and biocide resistance genes, aggregate

in a process that is often reliant on the activity of several key insertion elements and homologous recombination (26–28). The structures of class 1 integrons continue to be shaped by antimicrobial selective pressures and the activity of insertion elements such as IS*26* (29–31).

Here, we present whole-genome sequencing (WGS) data derived from a large study (*n* = 425) of MDR *E. coli* isolates sampled from Australian silver gulls. The isolates were selected based on phenotypic nonsusceptibility to three clinically important antibiotics and were derived from cloacal swabs of gull chicks inhabiting three sites (Five Islands [FI], Montague Island [MI], and White Bay [WB]) on or near the coast of New South Wales, Australia, in 2012. Montague Island is a nature reserve geographically distant from major Australian cities and is home to more than 90 bird species. A comprehensive AMR phenotype and phylogenomic analysis of *E. coli* and mobile genetic elements involved in the dissemination of antimicrobial and virulence genes is reported, with AMR biomarkers that can be validated for their utility as epidemiological markers to track MGEs carrying combinations of clinically significant resistance genes. For the first time, we provide a graphical display of antibiotic resistance genes, virulence genes, and F virulence plasmid carriage with linkage to phylogeny and show that while there is a striking diversity of *E. coli* lineages that carry a high antibiotic gene load, only select lineages carried multiple genes encoding virulence and AMR.

## RESULTS

**Extreme diversity in *Escherichia coli* isolates hosting clinical antibiotic resistance in an urban avian setting.** *E. coli* isolates (425 isolates) from the cloacae of 504 gull chicks from two human-impacted sites (Five Islands Nature Reserve [FI] and White Bay [WB]) and one potentially pristine reserve, Montague Island (MI), all in New South Wales, Australia, were recovered on media supplemented with antibiotics critical for the treatment of human infections. Reduced susceptibility was identified to cefotaxime (273 isolates [144 from FI, 86 from WB, and 43 from MI]), meropenem (38 isolates, all from FI), and ciprofloxacin (115 isolates [48 from FI, 28 from WB, and 39 from MI]). Notably, isolates carrying key carbapenem resistance genes were also recovered on cefotaxime (28 isolates)- and ciprofloxacin (14 isolates)-supplemented agar plates, also exclusively from FI.

Short-read whole-genome sequences for 424 isolates, referred to as the collection here, were assembled to examine their population structure. The collection contained 96 sequence types (STs), represented all the major *E. coli* phylogroups, and was predominantly comprised of commensal phylogroups A and B1. The collection also included examples of nearly clonal populations (e.g., ST1139); known ExPEC lineages such as ST38, ST58, ST69, and ST131 (32); emerging *E. coli* pathogens of ST457 (12) and ST216 (11); and a range of multiple-antibiotic-resistant "commensal" lineages that are also potential ExPEC lineages (e.g., ST10 [33]). Isolate summaries are provided in Table S1 in the supplemental material.

**Trackable antimicrobial resistance and virulence traits are associated with a range of mobile genetic elements within the *Escherichia coli* distribution.** Phylogenetic analysis depicting the distribution of *E. coli* lineages within the collection is shown in Fig. 1 and 2. The structure of the tree highlighted key split points between the *E. coli* phylogroups (cryptic IV [*n* = 1], B2 [*n* = 19], D [*n* = 60], F [*n* = 79], E [*n* = 6], C [*n* = 8], B1 [*n* = 101], and A [*n* = 150]) in midpoint-rooted distance order) and grouped subclades by sequence type and serotype.

**Class 1 integrons carrying clinical resistance genes are heavily impacted by IS*26* activity.** Two methods, read mapping and BLASTn alignments to the WGS assemblies, were used to interrogate the collection for class 1 integrase genes (*intI1*) as insertion sequence (IS) elements are eroding the integrity of the integrase by generating deletions at the 3′ end in Australia (34, 35) and internationally (36). The read mapping method identified *intI1* in 176 isolates, associated with the sulfonamide resistance genes *sul1* (*n* = 151), *sul2* (*n* = 169), and *sul3* (*n* = 16). Notably, the macrolide resistance gene *mphA* was identified in 114 isolates, most of which (109/114; 96%) simultaneously carried a copy of *sul1*. There was also evidence in gulls where the 3′-conserved

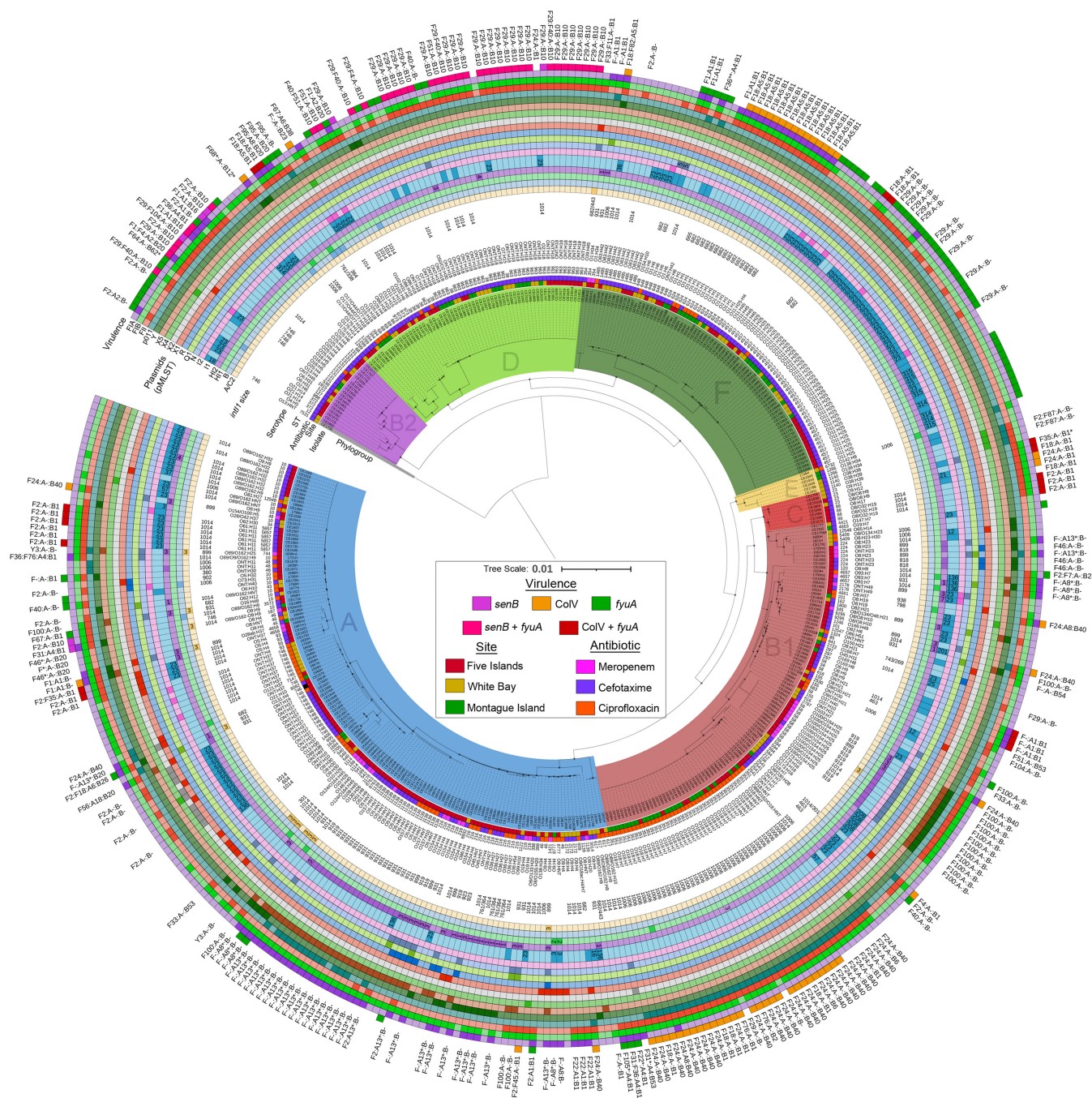

**FIG 1** Midpoint-rooted phylogeny of 424 *Escherichia coli* isolates from Australian silver gulls. The phylogenetic tree (PhyloSift) shows the distribution of short-read whole-genome sequences of *Escherichia coli* from this study. Presented around the tree are isolate metadata, including phylogroup, geographic site, antibiotic used for selection, sequence type, serotype, BLASTn match sizes to the class 1 integrase *intI1*, plasmid typing data, and pMLST if available. The key virulence content is identified in colored boxes along with F plasmid RST data on the edge. Intense color indicates a positive hit within the plasmid typing rings. The tree scale is presented in substitutions per site.

segments (CSs) of class 1 integrons were modified by the action of insertion elements, as has been described previously (24, 27, 29). Twenty-five isolates carried *intI1* and *floR*, a resistance gene for florfenicol, an antibiotic used to treat animal infections. Together, these gene combinations are indicative of specific integron subtypes.

Using the BLASTn approach, however, *intI1* was identified in 245 isolates, 57% of our collection, with a notable range of consistent truncation sizes. Twelve *intI1* hit sizes (having excluded any under 250 bp) that arise from confirmed truncations were identified, plus seven more that stem from scaffolding in the whole-genome sequence assemblies.

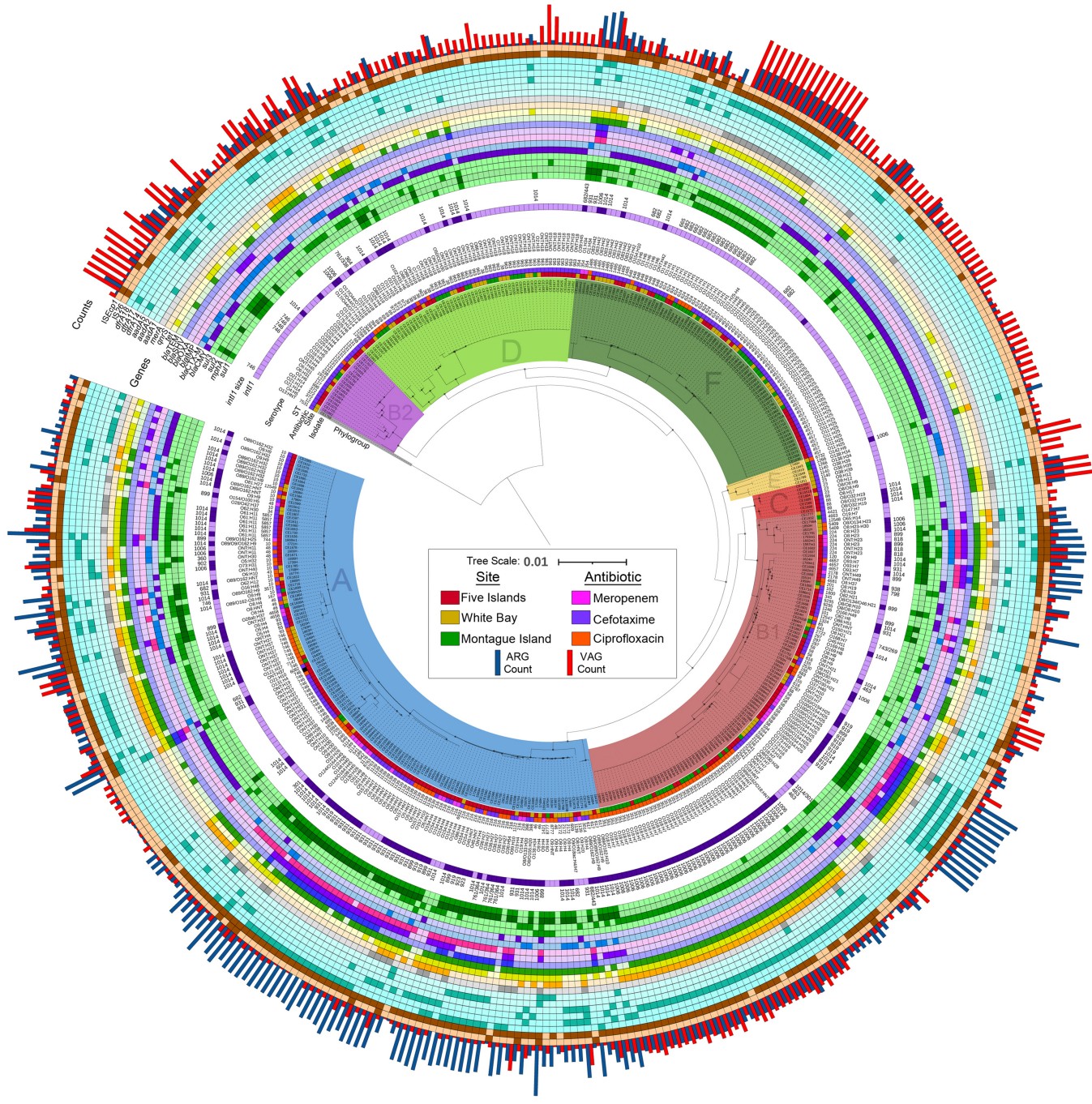

**FIG 2** Midpoint-rooted phylogeny of 424 *Escherichia coli* isolates from Australian silver gulls aligned to genotype data. The phylogenetic tree (PhyloSift) shows the distribution of short-read whole-genome sequences of *Escherichia coli* from this study. Presented around the tree are isolate metadata, including phylogroup, site, antibiotic used for isolation, sequence type, serotype, BLASTn match sizes to the gene *intI1*, and AMR genotyping data. Darker colors indicate positive hits within the genotyping rings. The tree scale is presented in substitutions per site. The histogram on the outer edge shows the ARG count (blue) (range, 0 to 24) and virulence-associated gene (VAG) count (red) (range, 0 to 38) for each whole-genome sequence.

The latter, while not as useful for making epidemiological linkages to external data sets, served to characterize some identical structures within this collection. The truncations were almost exclusively IS*26* mediated and targeted the 3′ end of *intI1*, hence not disrupting the cassette promoter but eliminating integrase activity. Annotations for representative contigs hosting these Δ*intI1* genes are presented alongside seven annotations for integrons hosting complete *intI1* genes in Table S2. Hit sizes for *intI1* are also presented alongside plasmid data (Fig. 1) and genotype data (Fig. 2).

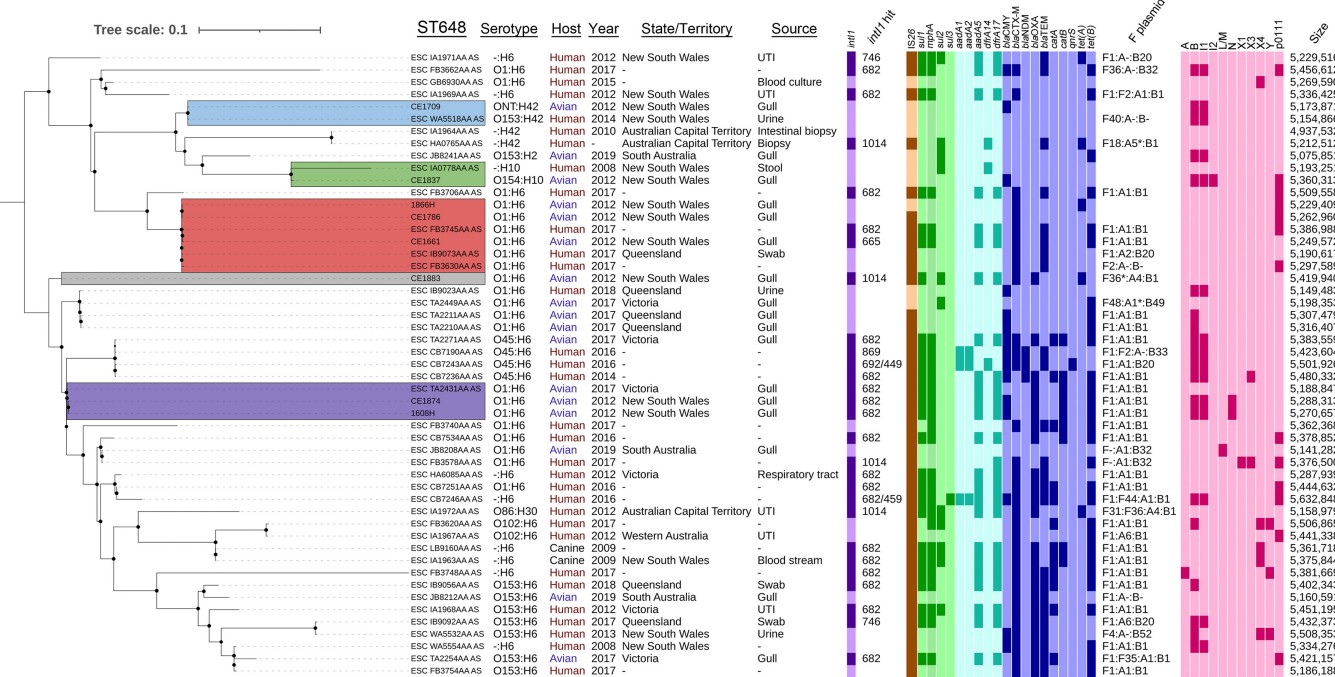

**FIG 3** Phylogenetic distribution of Australian *Escherichia coli* ST648 isolates with associated metadata and genotyping. Clades including isolates from this study are colored. Darker colors represent positive hits within the genotyping data. The tree scale is presented as SNVs per site. The single nucleotide polymorphism (SNP) analysis spans an average of 70% of each whole-genome sequence. UTI, urinary tract infection.

Significant insights into understanding the potential human health threat posed by each of the major ST lineages became evident after overlaying critical virulence and antibiotic resistance genes carried by each isolate (Fig. 1 and 2). Comprehensive genotype profiles detailing the abundance and distribution of ARGs, VAGs, plasmid replicons, and ISs are catalogued in Table S1. Notably, $bla_{IMP-4}$ and $bla_{SHV-12}$ (outside a single isolate carrying $bla_{SHV-12}$) were identified only in isolates sourced from gull chicks at the Five Islands site.

Where detectable using BLASTn, each *sul3*-carrying isolate hosted a $\Delta mefB_{260}$ signature, the most common deletion signature observed in *sul3*-associated integron structures (37). A novel $\Delta mefB_2$ signature was also identified. Using this approach, we were able to evaluate how deletion variants partition according to sequence type and plasmid type. For example, by comparing isolates carrying $bla_{IMP-4}$, it highlights the elements repeatedly involved in its distribution and a clear association with clinically important multiple-drug resistance (Table S3).

**Evidence of interspecies transmission through exploration of a nationally distributed lineage.** The ST648 (*n* = 8) isolates in our collection were identified as O1:H6 (except for one ONT:H42 isolate) and notable for the diversity of carriage of class 1 integrase deletions ($\Delta intl1$) while carrying essentially the same resistance gene cargo (*dfrA17-aadA5-sul1-mphA*). Notably, the resistance cargo coaligned with different F plasmid RSTs, including F1:A1:B1 ($\Delta intl1_{682}$ and $\Delta intl1_{665}$ signatures) and F36*:A4:B1 (complete *intl1* gene; 1,014 bp). This sequence type was of additional interest due to the occurrence of Australian avian and human ST648 isolates in EnteroBase. A phylogenomic analysis, which included 43 Australian ST648 reference genome sequences with appropriate metadata, identified several sublineages of O1:H6 isolates, an O45:H6 lineage, and a cluster of O153:H42 isolates from humans and gulls (Fig. 3). A class 1 integrase gene was detected in 28 ST648 isolates, and 19 (68%) of these carry the $\Delta intl1_{682}$ signature (seemingly) on F1:A1:B1 plasmids. Several isolates carry a full copy of *intl1* (1,014 bp; 4 isolates; multiple F RSTs), $\Delta intl1_{746}$ (2 isolates; F1:A1/A6:B20 associated), $\Delta intl1_{665}$ (1 isolate; F1:A1:B1), $\Delta intl1_{869}$ (1 isolate), and $\Delta intl1_{962}$ (1 isolate), highlighting the potential of using *intl1* deletions as epidemiological markers to track bacteria and

the mobile elements that carry these markers (Fig. 3). Two human-sourced isolates (IA1971AA and IB9092AA) from the EnteroBase data set were positive for *senB*, critically in combination with a $\Delta intI1_{746}$ signature that also correlates with ST131, ST12, and ST10 isolates observed in our collection.

**Carbapenemase-encoding plasmids of subtype HI2 pST3 are important distributors of multiple-drug resistance.** In our collection, isolates simultaneously carrying HI2 plasmid sequence type 3 (pST3) plasmids, $\Delta intI1_{931}$ signatures (among others [Table S2]), and the $\beta$-lactamases $bla_{IMP-4}$ and $bla_{OXA-1}$ were prevalent. The complete sequences of several HI2 pST3 plasmids were resolved using long-read sequencing, including pCE1681_A (ST216; HI2 pST3), p1566m1 (ST58 O100/O154:H25; HI2 ST3), and p1585m1_A (ST58 O100/O154:H25; HI2 nontypeable). Note that it was common for HI2 plasmid subtyping to fail when the signature $\Delta intI1_{931}$ was present, particularly in ST58 and ST1139 isolates; however, HI2 plasmid replicons were detected in these strains. Plasmid p1585m1_A was included to represent this mistyping. Comparisons of the annotated plasmid sequences and phylogenetic analyses were used to test the similarity of these pST3 plasmids to each other and to reference HI2 plasmids belonging to HI2 pST1 and pST3 (Fig. 4A and C). The HI2 pST3 plasmids sourced from Australian silver gulls grouped distantly from other HI2 pST3 plasmids (Fig. 4B). An analysis of variant sites attributed this segregation to a set of over 100 single nucleotide variants (SNVs) that were confined to the *ter* region in these plasmids. Reanalysis of HI2 pST3 plasmids with the *ter* region excluded (approximately 10 kb) placed the plasmids among a clade of three Australian HI2 pST3 plasmids of porcine origin, in a cluster separated by approximately 25 SNVs (data not shown). This suggests that an endemic Australian lineage of HI2 pST3 plasmids exists, mobilizing AMR in multiple settings, and that the *ter* operon(s) may be utilized to distinguish future plasmids (25, 38).

An alignment (Fig. 4C) of the Australian gull HI2 pST3 plasmid lineage with pSTM6-275 (GenBank accession number CP019647.1) (39), a representative plasmid isolated from *Salmonella enterica* in an Australian commercial swine operation, demonstrated a consistent plasmid structure with the acquired ARGs and MGEs linked to similar sites in the backbone. In contrast, two separate regions of the plasmid (~100 kb) comprising the plasmid backbone and acquired cargo were absent from the nontypeable HI2 plasmid of ST58. Annotation of two class 1 integron structures in the HI2 pST3 plasmids resolved here (Fig. 4A) determined that the cassette contents of integron structures in p1566m1_A were almost identical to those of the integron hosting $bla_{IMP-4}$ in an HI2 ST1 plasmid in Australian *Salmonella enterica* serovar Typhimurium from cats (40). Points of difference included the following: (i) the genes *aacA4*, *catB3*, and *arr3* remained present but were linked here to the $\Delta intI1_{931}$/*mphA* class 1 integron structure alongside $bla_{OXA-1}$; (ii) $bla_{IMP-4}$ appeared as a lone cassette in a class 1 integron structure bounded by IS*26* elements in a distinct insertion point near *tetAR*; (iii) an N plasmid *rep* gene had been captured on these plasmids, which can be seen in the correlation between HI2 and N replicon typing from our collection; (iv) the class 1 integron resolved within the nontypeable HI2 plasmid had cassettes in the order *intI1*-$bla_{IMP-4}$-*qacG*-*aacA4*-*catB3*, matching the gene organization reported previously for the HI2 ST1 plasmid (11), although here, it did not include the *mphA*-associated region; and (v) from genotyping, it appears that the presence of *mer* was highly variable on these plasmids (Fig. 2). These observations point to ongoing evolution shaping these important drug resistance plasmids.

**F plasmid subtypes mobilizing key uropathogen virulence genes.** Using the criteria described previously by Liu et al. (17), 18% (*n* = 74) of our collection carried a ColV F virulence plasmid. This remarkably high level of carriage warrants further scrutiny, with at least 20 sequence types being implicated in carriage. To determine the presence of known virulence regions and an F plasmid backbone, we compared representative isolates from each sequence type and RST combination against plasmid pSDJ2009-52F (GenBank accession number MH195200.1), an F2:A−B1 ColV plasmid taken from a clinical Australian *E. coli* ST58 isolate (24) (Fig. 5). All but the F1:A1:B−

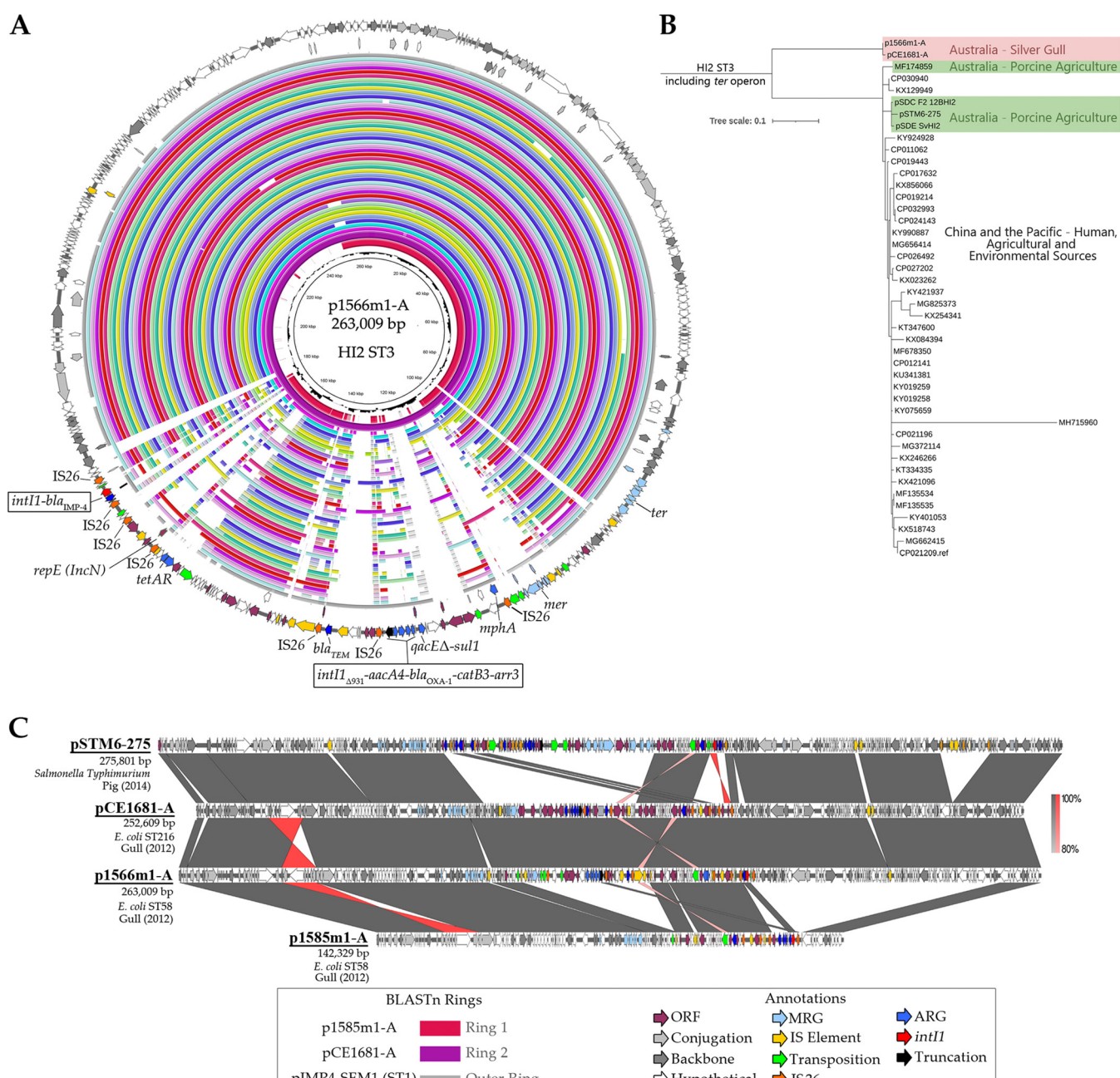

**FIG 4** Analysis of HI2 pST3 plasmids and comparison to pST1 plasmid pIMP4-SEM1. (A) The annotation of p1566m1-A is presented on the outer ring, with an internal BRIG alignment of all other HI2 pST3 plasmids plus pIMP4-SEM1. (B) SNV phylogeny of pST3 plasmid sequences. (C) Linear annotated diagrams of key HI2 plasmids coupled with BLASTn alignment data visualized using EasyFig. Note that the short-read whole-genome sequence of isolate 1566m1 was excluded from other analyses due to quality cutoffs. ORF, open reading frame.

representative showed high coverage of the reference plasmid backbone and virulence regions.

The F2:A⁻B1 representatives were distinctly similar to the clinically sourced reference plasmid, highlighting a concerning connection between microbiomes. Additionally, investigation into the AMR region of isolate CE1572 (ST88) revealed an interesting distinction not clearly discernible from the otherwise highly similar visual comparison presented in Fig. 5. These plasmids host an unusual CRR distinguished by a Tn*21*/Tn*1721* fusion transposon, a class 1 integron hosting *dfrA5*, and several IS*26*-mediated structures hosting *bla*<sub>TEM</sub>, *sul2*, and *strAB* in a Tn*6029* structure (24). Tn*6029* and variants of it have a history of association with plasmids found in zoonotic pathogens (27, 41–43). The ST88 lineage,

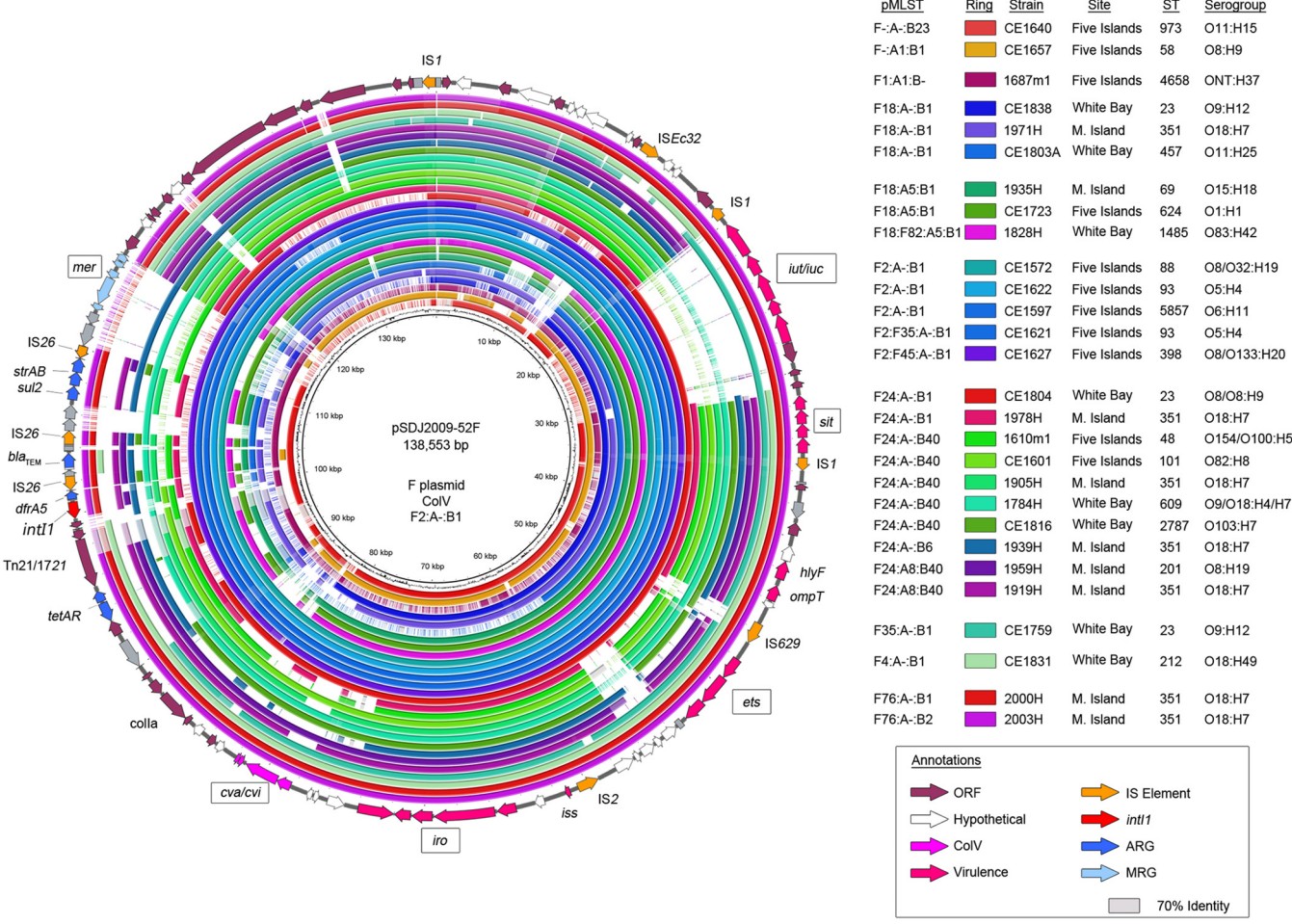

**FIG 5** Alignment of ColV-positive isolates against plasmid pSDJ2009-52F. The annotation of pSDJ2009-52F is presented around a BRIG alignment of representative whole-genome sequences containing ColV plasmid genotypes. Rings are grouped by F RST.

represented by CE1572, was notable in that it still carried *sul1*, unlike in pSDJ2009-52F, where it had been removed by IS*26* activity, suggesting that it is a progenitor of the clinical structure.

Using the *senB* toxin gene as a proxy for identifying isolates carrying pUTI89-like F virulence plasmids, we aligned *senB*-positive isolates of various sequence types in this collection, plus two ST648 isolates from EnteroBase notably sharing key gene signatures, against reference plasmid pUTI89 (GenBank accession number CP000244.1) (Fig. 6). Outside singular exceptions of apparent DNA loss, plasmids of subtypes F1, F2, F29, F31, F36, and F51 all appeared to carry the known virulence genes found on pUTI89, with notably high synteny shown between the reference backbone and isolates hosting both F2 and F29 subtype plasmids. The loss of alignment coverage at IS*26* in many of the isolates indicates potential AMR gene capture, a notable observation given that diverse *E. coli* isolates that carry pUTI89 are often pansusceptible and carry restricted plasmid contents (44).

**Antibiotic resistance phenotypes were concordant with genotypes.** Phenotypic resistance was determined against 16 antibiotics (Table S1), with the highest rates of resistance belonging to ampicillin (99.3%) and cephalothin (93.2%). Phenotypic resistance to streptomycin (58.0%), sulfonamides (57.3%), sulfamethoxazole-trimethoprim (52.6%), and tetracycline (50.9%) was also observed. With the exception of colistin (only 2 isolates), all tested antibiotics had a resistance rate of >16% (i.e., a minimum of 70/425 isolates were phenotypically resistant to each antibiotic), an observation that attests to the MDR status of *E. coli* inhabiting the gastrointestinal tract of silver gulls.

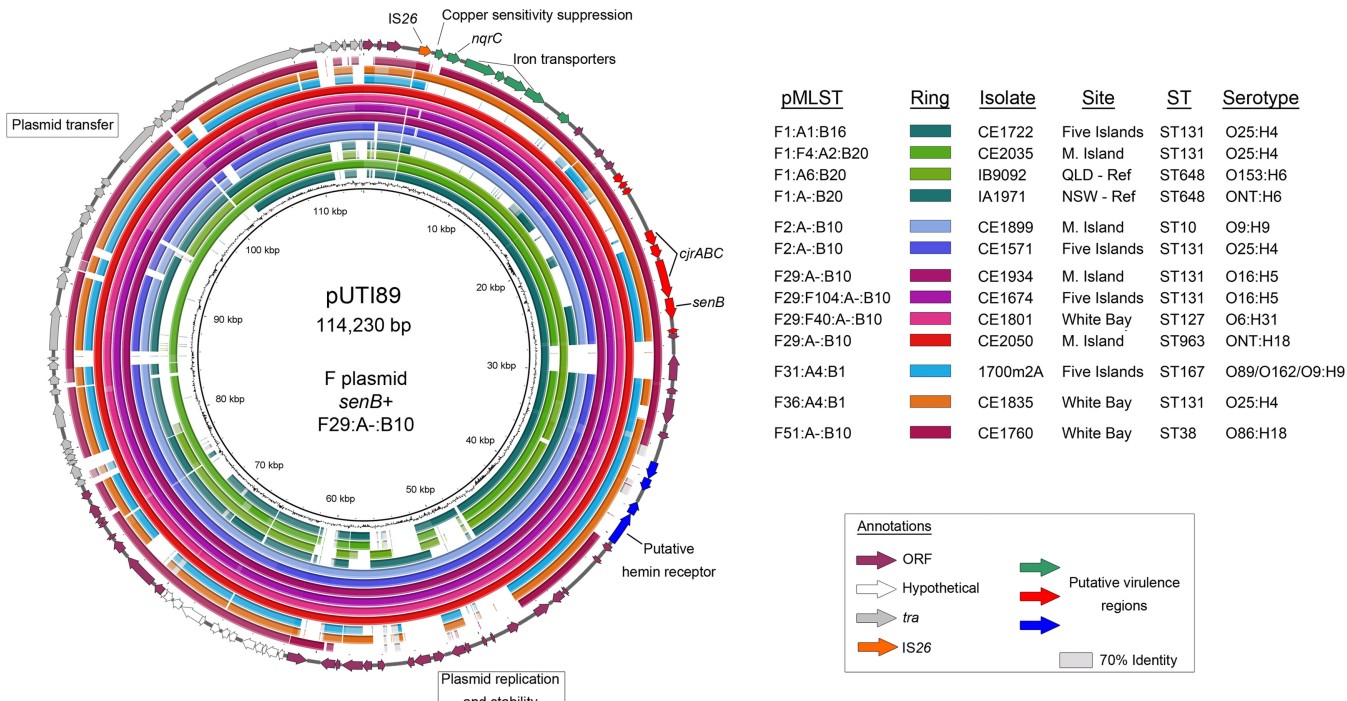

**FIG 6** Alignment of *senB*-positive isolates against plasmid pUTI89. The annotation of pUTI89 is presented around a BRIG alignment of *senB*-positive whole-genome sequences. Two ST648 reference sequences were included.

Except for streptomycin (discordance of 32.8%), there was concordance between phenotypic antimicrobial susceptibility (AS) testing and WGS-based AS prediction (Table S4). The discordance for streptomycin was due to "intermediate" phenotypes with no gene detection and was primarily associated with isolates that carry $bla_{CMY}$ in 103 of the 127 isolates with intermediate phenotypes.

Chloramphenicol, with 7.55% discordance, was associated with the detection of *catB* family genes in isolates that were phenotypically susceptible. The only other antimicrobial tested with a genotype/phenotype discordance rate higher than 5% (6.13%) was ceftazidime. In this case, it appeared that $bla_{CTX-M-14}$ (12/14 isolates) was not conferring its expected resistance. Regarding colistin, only two isolates demonstrated resistance, one of which was positive for *mcr-1*. Only potential mutations in *pmrAB* explained resistance in the other. Unknown mutations were also used to provide a potential reason for resistance in cases of intermediate or resistant phenotypes to nalidixic acid. We had 22 such isolates and found mutations in quinolone resistance-determining regions (QRDRs) in each. Interestingly, we frequently detected a *parC* E62K mutation, with it being the only detected mutation in 10 isolates.

The results of Mastdiscs corresponded well with the phenotype unless the carbapenemase genes, including $bla_{OXA-1}$ or $bla_{OXA-10}$ alleles, were present. The presence of these β-lactamases resulted in differing results across the collection. Of the remaining 252 tested isolates, phenotypic results of 249 correspond with the predicted AMR from WGS. Phenotypic data are presented against the collection phylogeny in Fig. S1.

**Final assessment.** Across the collection, lineages either were often multiple-drug resistant or carried a heavy virulence gene load. This observation suggests that lineages carrying both were uncommon at the time of isolation, but examples like ST624 and ST38 were exceptions. The general trend of isolates favoring heavy AMR carriage or heavy virulence carriage indicates that multiple-drug resistance was often mobilized by different elements to virulence genes, with HI2 pST3 plasmids being responsible for the most severe AMR carriage. The genes $bla_{IMP-4}$ and $bla_{SHV-12}$ were centered around the Five Islands site, indicating that anthropogenic contact could play a role in AMR gene persistence for these critical resistances.

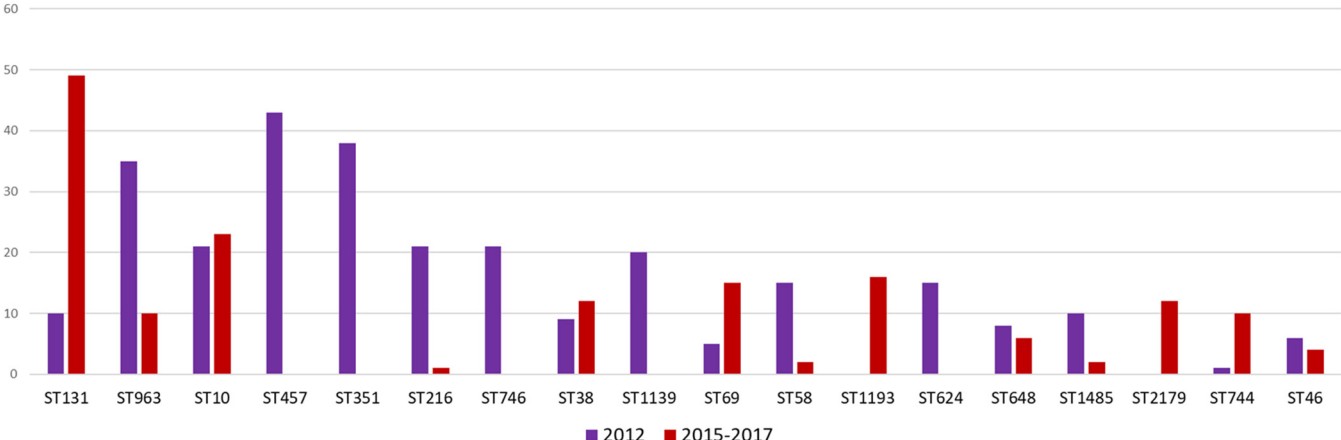

**FIG 7** Histogram of counts of sequence types from studies of *E. coli* in Australian silver gulls in 2012 and from 2015 to 2017. Included are sequence types with more than 10 isolates total from either study.

## DISCUSSION

Here, we demonstrate that bird populations adapted to forage from landfills, despite never receiving antimicrobials to treat infection, carry a full spectrum of highly and critically antibiotic-resistant *E. coli* lineages. Among others, ST10, ST648, ST351, ST1139, ST457, ST216, ST963, ST93, ST23, ST88, and ST5857 should be monitored as potential emerging human pathogens due to the capture of critical ARGs and virulence gene cargo, potentially on ColV and pUTI89-like F virulence plasmids. Studies of *E. coli* ST457 and ST216 showing evidence of interspecies transmission and carriage of drug resistance plasmids have been reported (11, 12), and ST58 has been flagged as a sequence type of emerging significance (24, 45). Despite the distance separating the sampled sites along the New South Wales coast, the transfer of lineages between the separate breeding populations is suggested based on the detection of isolates of the same ST and serotype and hosting the same mobile genetic elements (e.g., ST624, ST457, or ST351). A notable exception, however, is meropenem resistance, mediated by the $bla_{\text{IMP}}$ genes, which was sourced exclusively from Five Islands. Based on previous observations of gull diet at these sites, it is likely that the Five Islands birds have the most frequent contact with contaminated food sources, in proximity to a municipal sewage plant and landfill sites (46).

A previous study used fecal samples taken from adult gulls from 2015 to 2017 from across Australia (9). A comparison of the *E. coli* sequence types isolated in both studies demonstrates the persistence of numerous sequence types at the three New South Wales sites and elsewhere but also demonstrates potential site-specific clusters (Fig. 7). Across the two studies, a total of 30 sequence types were found to be conserved, approximately one-third of the total sequence type diversity in either study. Whether this infers that some drug-resistant *E. coli* isolates are only transient colonizers of the gull microbiome remains to be determined. Previous work has demonstrated that *Salmonella* species can persist in the soil of gull breeding sites (47), suggesting that certain sites could become hot spots for certain microbes and MGEs.

We provide substantial evidence from these data of the diversity and abundance of atypical integrons where the class 1 integrase is subjected to multiple IS-mediated deletions as well as IS-mediated decay of the 3′-CS, providing a sage reminder to exercise caution when setting cutoffs while interrogating genomic data. The conspicuously low abundance of *merA*, in comparison to *intI1* and *sul1* particularly, also points to the ongoing evolution of CRRs. Multiple-drug resistance was common among the critically resistant isolates within our collection and strongly correlated with the presence and diversity of class 1 integrons. The most common integron structures disseminated internationally are associated with individual *sul* genes (primarily *sul1*, *sul2*, or *sul3*) (29, 48, 49).

The most dominant type of integron in our collection was the *sul1*-associated class 1 integron hosting the macrolide inactivation gene cluster, defined by the capture of a region carrying IS*6100*-*mphR*-*mrx*-*mphA*. This dominant integron type has several notable characteristics and has been reported in clinical settings in Australia (34, 35, 50).

The most intriguing meropenem resistance vectors identified were HI2 pST3 plasmids. These plasmids were significant both for the range of *E. coli* sequence types that harbored them and for the diversity of *intI1* variants associated with them. This plasmid subtype is a noted vector for antimicrobial resistance genes, mostly in the Asia-Pacific region (25, 51, 52). In Australia, HI2 pST3 plasmids have been reported to mobilize a range of resistance in both *E. coli* (38) and *Salmonella enterica* serovar Typhimurium (25) in agricultural settings. Here, we report HI2 pST3 plasmids acquiring known class 1 integron structures hosting $bla_{IMP-4}$ and the further capture of $bla_{OXA-1}$. Phylogenomic analyses of these plasmids also revealed a set of variant sites within the *ter* region of the HI2 plasmid backbone that can serve to distinguish the plasmid lineage resolved here from other plasmids in the future.

Many molecular signatures, most of which are likely the product of deletion events caused by insertion elements, particularly IS*26*, were also identified here. These truncations were notable for the fact that they do not affect the cassette promoter but have implications for cassette shuffling via the loss of integrase activity. Tracking these deleted sequences may have merit from an epidemiological standpoint and highlights potential pitfalls in using PCR-based methods to detect *intI1* and class 1 integrons more broadly. This was highlighted particularly by associations between $bla_{IMP-4}$ and $intI1_{\Delta931}$ and $qnrS1$/$intI1_{\Delta1006}$, which were responsible for numerous meropenem- and ciprofloxacin-resistant isolates, respectively. Isolates hosting $intI1_{\Delta931}$ demonstrated the broadest range of phenotypic resistance. Our work highlights pitfalls in standard automated annotation and gene detection pipelines in being able to reliably characterize CRRs.

In our *E. coli* collection, different combinations of virulence genes were carried by different lineages, with some examples like ST648 (group F) being notably poor in recognized virulence potential. The F plasmids designated ColV and pUTI89-like/*senB* positive are key components for the success of several extraintestinal pathogenic *E. coli* lineages, particularly within the classically pathogenic phylogroups B2 and D/F, with additional considerations for the cocarriage of iron uptake systems represented by *irp2* and *fyuA*, which were consistently cocarried within the collection (53). The virulence content of these plasmids includes genes for iron acquisition, serum resistance, and biofilm formation, key attributes needed to colonize host gastrointestinal and extraintestinal sites (54). With the unusually high carriage of ColV plasmids within phylogroups B1 and C as well as the rare carriage of both plasmid types in phylogroup A, there is a broad dissemination of pathogenic potential among critically drug-resistant *E. coli* isolates hosted by Australian silver gulls. Although these ST648 isolates were poor in recognized virulence potential, locally sourced whole-genome sequences suggest a propensity to cause extraintestinal disease in humans and the carriage of similar CRRs.

The identification of ColV and pUTI89-like plasmids in potential ExPEC lineages inhabiting the Australian silver gull populations was a major observation in this study. pUTI89-like virulence plasmids are not known for their ability to carry AMR genes and are typically associated with lineages of *E. coli* that are pansensitive (44). However, pUTI89-like plasmids carry a copy of IS*26*, an insertion element noted for mobilizing antibiotic resistance genes and forming CRRs. Our analyses indicate that pUTI89 and variants of it carry antibiotic resistance genes (Fig. 6). It remains premature to speculate if the gastrointestinal tracts and feeding behaviors of gulls overtly influence plasmid and genome evolution. A limitation of our study is that we biased our sample collection toward isolates that display resistance to clinically important antibiotics. Studies are needed to determine if carriage rates of virulence attributes in *E. coli* isolated without an antibiotic are comparable. Further studies should seek to understand which *E. coli* lineages are natural colonizers of urban-adapted birds, whether drug-resistant and human-pathogenic *E. coli* lineages persist during the life span of individual birds, and

if birds play a significant role in advancing the evolution of virulence and antibiotic resistance in clinically important enterobacterial populations.

In conclusion, high-resolution genomic methods were used to examine the diversity of *E. coli* lineages resistant to critically important antimicrobials from Australian silver gulls living in human-associated and natural environments. Genomic diversity was a feature of the study not only from a chromosomal perspective but also in resistance and virulence gene profiles, integron structures, and plasmid lineages. Our analysis provides indications that *E. coli* isolates—resistant to critically important antimicrobials and living in gulls—frequently harbor virulence factors and are often equipped with a rich toolkit for horizontal gene transfer. Taken together, our results support the hypothesis that gulls and likely other urban wildlife visit anthropogenic sites contaminated by antibiotic residues and drug-resistant microbial populations, become colonized, and transmit these diverse *E. coli* lineages. The diversity of integron signatures points to evidence of an environment in the gull microbiome that supports the ongoing evolution of genetic vectors that harbor drug resistance and virulence genes.

## MATERIALS AND METHODS

**Sampling and strain isolation.** The sampling of *Enterobacteriaceae* from Australian silver gull samples in 2012 has been described previously (8). Briefly, cloacal samples ($n = 504$) were collected from gull chicks at three sites in New South Wales, Australia. Cloacal samples were enriched in buffered peptone water and then cultured on three MacConkey agar plates supplemented with one of the following clinically important antibiotics (CIAs): cefotaxime (2 mg/L), meropenem (0.125 mg/L), or ciprofloxacin (0.05 mg/L). From these, a total of 425 *Escherichia coli* isolates were obtained. Regarding the sites, Five Islands Nature Reserve (FI) ($n = 231$) (34°29′S, 150°56′E) sits off the coast of Port Kembla, 10 km from the city of Wollongong (population, 295,000), and hosts a breeding colony known to feed on human refuse almost exclusively. White Bay in Sydney (WB) ($n = 114$) (33°86′S, 151°18′E) acts as a second human-associated site. Montague Island (MI) ($n = 80$) (36°15′S, 150°14′E) is a nature reserve about 9 km from Narooma, a small village on the South Coast of NSW home to about 3,300 people.

**Phenotypic antimicrobial susceptibility testing.** A total of 424 strains were tested for susceptibility to a set of 15 antimicrobials (see Table S5 in the supplemental material) using the disk diffusion method on Mueller-Hinton agar (Oxoid, UK) with the *E. coli* ATCC 25922 control strain. Nonsusceptibility was determined according to Clinical and Laboratory Standards Institute (CLSI) guidelines (55, 56). Colispot tests were used to determine susceptibility to colistin as disk diffusion is not a suitable method for this antibiotic (57). A Mastdiscs Combi test for the evaluation of AmpC and extended-spectrum-$\beta$-lactamase (ESBL) production was performed for a total of 312 isolates, including all isolates from media containing cefotaxime and selected isolates from media with ciprofloxacin where the ESBL/AmpC phenotype was expected according to preliminary PCR screening.

**DNA isolation and whole-genome sequencing.** Genomic DNA for short-read sequencing was isolated using the NucleoSpin tissue kit (Macherey-Nagel GmbH & Co., Düren, Germany) according to the manufacturer's protocols. DNA libraries were prepared using a Nextera XT DNA sample preparation kit with modifications (37) and sequenced on a NovaSeq platform (Illumina, San Diego, CA, USA), resulting in 424 successful *E. coli* whole-genome sequences. Genomic DNA of two *E. coli* ST58 isolates was extracted using a NucleoSpin microbial DNA kit (Macherey-Nagel) and used to prepare DNA libraries according to standard Pacific Biosciences (PacBio) protocols. Sequencing was performed on a Sequel platform (PacBio, USA). Only one of these isolates (1585m1) is also represented in the short-read assemblies and phenotypic testing.

**Genotyping.** Detection of individual genes and sequence types and *in silico* serotype prediction determination were performed using ARIBA (58) in combination with ARIBAlord (https://github.com/maxlcummins/pipelord/tree/master/aribalord). The following gene databases were included: PlasmidFinder (59), ResFinder, and VirulenceFinder (60). Insertion sequences were identified with ISfinder (61). Plasmid multilocus sequence typing was performed using the Centre for Genomic Epidemiology server (59) (https://cge.cbs.dtu.dk/services/pMLST/). Phylogrouping was performed using the ClermontTyping server (62) (http://clermontyping.iame-research.center/). Point mutations conferring antimicrobial resistance were identified with PointFinder (63) through ResFinder (https://cge.cbs.dtu.dk/services/ResFinder/).

**Phylogenetics.** Sequences were confirmed to be *E. coli* using Kraken2 (64). A maximum likelihood phylogeny was constructed using PhyloSift (65) and FastTree 2 (66), using standard settings for both and the –gtr flag in FastTree 2 to use a general time-reversible model. Phylogenetic analyses of sublineages and plasmid sequences were performed using the Harvest suite (67). All visualization of phylogenetic trees was performed in iTOL (68). Reference whole-genome sequences were sourced from EnteroBase (69) (https://enterobase.warwick.ac.uk/species/index/ecoli). Reference plasmid sequences were sourced from GenBank (70) (https://www.ncbi.nlm.nih.gov/genbank/).

**Comparative genomics.** Whole-genome alignments were performed with progressiveMauve (71) for analytical purposes. Circular and linear visualizations of genome comparisons were generated with BRIG (72) and EasyFig (73), respectively. Annotations and their visualizations were handled using SnapGene (https://www.snapgene.com/). Automated annotations were generated by RASTtk (74).

**Data availability.** Sequences have been deposited under BioProject accession number PRJNA630096.

## SUPPLEMENTAL MATERIAL

Supplemental material is available online only.
**FIG S1**, TIF file, 14.2 MB.
**TABLE S1**, XLSX file, 0.5 MB.
**TABLE S2**, XLSX file, 0.01 MB.
**TABLE S3**, XLSX file, 0.04 MB.
**TABLE S4**, XLSX file, 0.01 MB.
**TABLE S5**, XLSX file, 0.01 MB.

## ACKNOWLEDGMENTS

We acknowledge the efforts of the Ithree Sequencing Facility at the University of Technology Sydney and the contributions of Kay Anantanawat. We also thank to Jarmila Lausova, Dana Cervinkova, Iva Kutilova, Eva Suchanova, Barbora Stefikova, Martina Masarikova, Jana Hofirkova, Katarina Stredanska, Petra Kocurova, Kristyna Martinkova, and Aneta Kovarova for their help in the laboratory. We thank Fiona MacIver for assistance with preparing and editing the manuscript.

E.R.W. performed genomic analyses, handled sequence data, and drafted and edited the manuscript. K.N. performed phenotypic resistance experiments and WGS concordance analysis, drafted the respective sections, and edited the manuscript. H.T. performed phenotypic resistance testing and genomic analyses. I.J. and I.B. performed antibiotic susceptibility testing and DNA isolation for WGS. S.P.D., M.D., and I.L. conceptualized the study, provided supervision, acquired funding, and edited the manuscript. All authors read and approved the final manuscript.

We declare that we have no competing interests.

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
