## [Reviewer comments · mSystems]

Urban wildlife crisis: Australian Silver Gull is a bystander host to widespread clinical antibiotic resistances

Ethan Wyrsh, Kristina Nesporova, Hassan Tarabai, Ivana Jamborova, Ibrahim Bitar, Ivan Literak, Monika Dolejska, and Steven Djordjevic

Corresponding Author(s): Steven Djordjevic, Australian Institute for Microbiology and Infection

Review Timeline:

Submission Date:

February 23, 2022

Accepted:

April 7, 2022

Editor: Holly Lutz

Reviewer(s): The reviewers have opted to remain anonymous.

Transaction Report:

DOI: <https://doi.org/10.1128/msystems.00158-22>

April 7, 2022

Prof. Steven Philip Djordjevic
Australian Institute for Microbiology and Infection
15 Broadway, University of Technology Sydney
Ultimo, NSW 2007
Australia

Re: mSystems00158-22 (Urban wildlife crisis: Australian Silver Gull is a bystander host to widespread clinical antibiotic resistances)

Dear Prof. Steven Philip Djordjevic:

Your manuscript has been accepted, and I am forwarding it to the ASM Journals Department for publication. For your reference, ASM Journals' address is given below. Before it can be scheduled for publication, your manuscript will be checked by the mSystems production staff to make sure that all elements meet the technical requirements for publication. They will contact you if anything needs to be revised before copyediting and production can begin. Otherwise, you will be notified when your proofs are ready to be viewed.

Publication Fees:

We recognize that the video files can become quite large, and so to avoid quality loss ASM suggests sending the video file via <https://www.wetransfer.com/>. When you have a final version of the video and the still ready to share, please send it to mSystems staff at mSystems@asmusa.org.

For mSystems research articles, if you would like to submit an image for consideration as the Featured Image for an issue, please contact mSystems staff at mSystems@asmusa.org.

Sincerely,

Holly Lutz
Editor, mSystems

Journals Department
Figure S1: Accept
Table S5: Accept
Table S3: Accept
Table S2: Accept
Table S4: Accept
Table S1: Accept